

# The regulatory effect of blood group on ferritin levels in aging: a retrospective study

Ni Xiaolin[1], Fenghong Yao[2] and Mingkang Pan[2]

[1] State Key Laboratory of Respiratory Health and Multimorbidity, Institute of Basic Medical Sciences, Chinese Academy of Medical Sciences and Peking Union Medical College, Beijing, China
[2] Beijing Zhongguancun Hospital, Beijing, China

## ABSTRACT

**Background**. Ferritin plays a pivotal role in the ageing process. Previous studies have identified statistically significant differences in ferritin levels among various ABO blood groups. However, the interaction between the ABO blood group and ferritin levels during senescence remains underexplored.

**Methods**. This research was conducted as a retrospective study involving a cohort of 3,843 individuals aged 40 and over who underwent blood type and ferritin testing at Beijing Zhongguancun Hospital. Assumption testing is employed to assess the normal distribution of continuous variables in the context of regression analysis. Spearman correlation analysis was employed to examine the relationship between the non-normally distributed biochemical indicators and ferritin levels. Age was considered the independent variable, while gender and biochemical indicators related to ferritin served as control variables. Blood type was analyzed as a moderating factor to evaluate its impact on the relationship between age and ferritin levels.

**Results**. Our findings revealed a negative correlation between ferritin and age ($\rho = -0.099$, $p < 0.001$). Significant differences in ferritin levels were observed between genders ($p = 0.005$) and blood groups ($p < 0.001$). The influence of age on ferritin levels varied across different blood groups, particularly in individuals with blood types A ($p = 0.003$, $\beta = -0.072$) and B ($p < 0.001$, $\beta = -0.110$), where the negative association between age and ferritin was more pronounced.

**Conclusion**. ABO blood type may influence ferritin levels as individuals age. Notably, in individuals with blood types A and B, the inverse relationship between age and ferritin levels was particularly significant among middle-aged and elderly individuals. These findings suggested the potential benefit of targeted iron supplementation for this population.

Corresponding authors
Ni Xiaolin, xiaolin_ni7@126.com
Mingkang Pan, panmkang@163.com

## INTRODUCTION

Rapid population ageing has emerged as a significant public health challenge in China and globally, as the deterioration of tissues and organs associated with ageing is a primary contributor to many chronic diseases (*Liu et al., 2018*). This degeneration will be reflected in the fluctuations of physiological indicators, which indirectly reflect the state of health
(*Dieteren et al., 2020*). As a result, it is imperative to implement preventive strategies and interventions that focus on critical physiological indicators conducive to healthy ageing.

Among the various physiological indicators, ferritin, which serves as a principal iron storage protein, is integral to numerous physiological and pathological processes. These include coronary artery disease, malignancies, sideroblastic anemias, neurodegenerative disorders, and hemophagocytic syndrome (*Li et al., 2024*). It has been proposed that serum ferritin (SF) can transport iron into cells, constituting the main pathway of iron supply to oligodendrocytes for myelin production, and then involved in neurotransmitter synthesis in the central nervous system, thus affecting working memory (*Rosell-Díaz et al., 2023*). In later life, reduced iron stores are associated with an increased risk of impaired physical and cognitive capacity, as well as up to a twofold higher risk of mortality (*Rosell-Díaz et al., 2023*; *Philip et al., 2020*). A cross-sectional study of the Ageing Health cohort (aged 65–90 years) found that SF positively correlates with neocortical amyloid-β load (NAL), suggesting that ferritin has the potential to serve as a blood biomarker panel for preclinical Alzheimer's disease (AD) (*Goozee et al., 2018*). In addition, iron plays a crucial role in erythropoiesis, oxygen transport throughout the circulatory system, mitochondrial respiration, and protection against free radicals in high-energy-demand cells, such as cardiomyocytes (*Hoes et al., 2018*). Research has demonstrated that serum ferritin levels decline with age, while the prevalence of heart failure (HF) rises (*Aboelsaad et al., 2024*). Therefore, understanding the characteristics of iron levels with age may have significant clinical implications, especially considering the availability of routine blood tests to detect iron deficiency and various therapeutic options for iron repletion.

Recently, an intriguing study revealed that statistically significant variations in ferritin levels across different ABO blood groups ($p < 0.001$) among a total of 7,723 healthy blood donors. Specifically, individuals with blood type A exhibited lower concentrations of ferritin, whereas those with blood type B displayed markedly elevated ferritin levels (*Franchini et al., 2016*). The relationship between serum ferritin levels and the incidence of COVID-19 is also influenced by ABO blood groups (*Lawaczeck, 2020*). As we know, ABO blood groups are ABO antigens (*i.e.,* A, B, AB, and O), which are glycoproteins and glycolipids expressed by the ABO gene on the surface of red blood cells, as well as a variety of human cells and tissues (*Anani et al., 2020*). Therefore, the clinical significance of ABO blood type extends beyond transfusion medicine and solid organ or hematopoietic transplantation. These antigens may also participate in the pathogenesis of various systemic diseases, including cancer, diabetes, infectious disorders, and cardiovascular disease (*Abegaz, 2021*). Therefore, the vast majority of studies have limited their investigation of this intriguing relationship to cohorts of high-risk patients, resulting in a scarcity of evidence regarding the physiological influence of ABO antigens on baseline levels of hematological and metabolic parameters. Given the changes in iron levels with age and the potential impact of blood type on these levels, it remains unclear whether blood type exerts a regulatory effect on iron levels in healthy ageing populations (Fig. 1). We propose that there exists an interaction between ABO blood group and ferritin levels throughout the aging process. To explore this ambiguous association, we conducted a study within the Beijing Health Examination Cohort (BHEC) to examine the influence of different blood

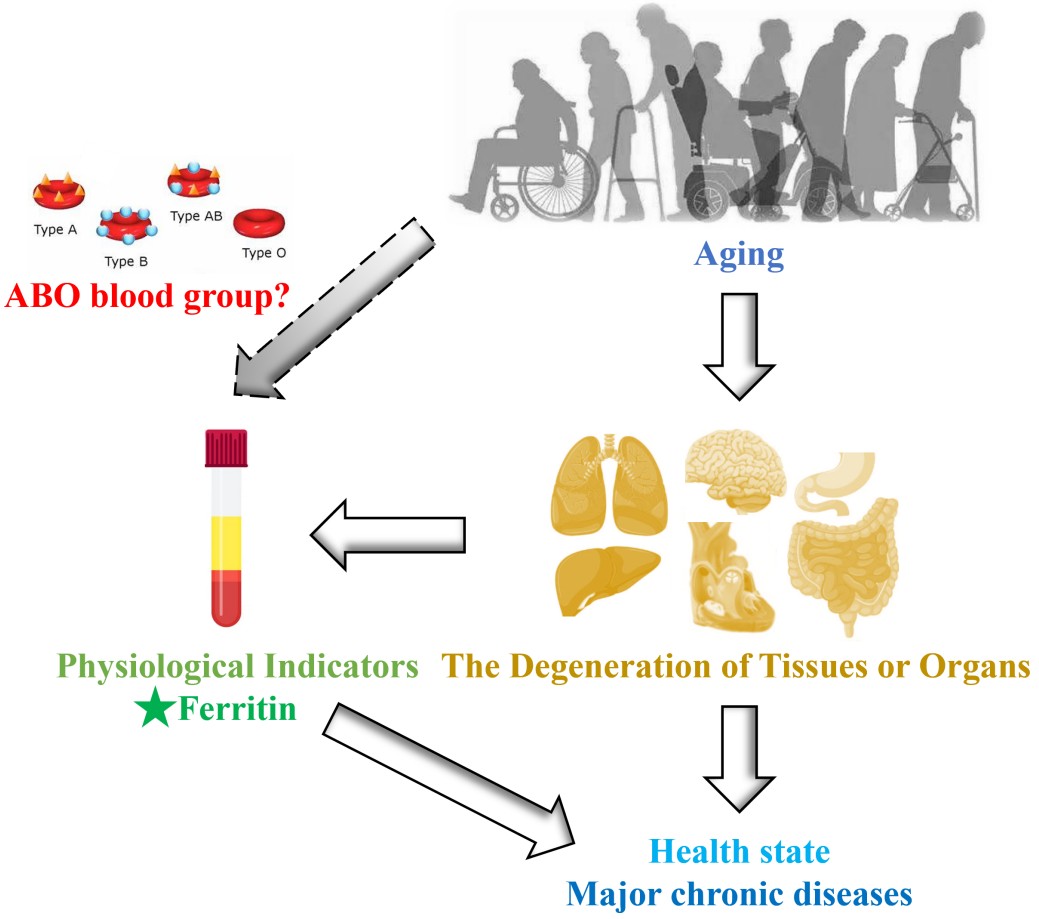

**Figure 1** Schematic diagram of this study design.

types on ferritin levels in ageing individuals, with the aim of guiding personalized clinical treatment in the future.

## MATERIALS & METHODS

### Study design and participants

The biochemical examination data were collected from 3,843 individuals aged 40 and above who underwent blood type and ferritin testing at Beijing Zhongguancun Hospital between July 2019 and July 2024. A total of 883 middle-aged and elderly individuals with missing a large number of examination data were excluded from the study. Ultimately, 2,960 healthy participants were included in this retrospective study. The Ethics Committee of the Institute of Basic Medical Sciences, Chinese Academy of Medical Sciences, approved the study protocol (ZS2024001). The study was conducted in accordance with the Declaration of Helsinki and its amendments. All participants provided written informed consent prior to enrollment. Clinical trial number: not applicable.

## Measures

Age, sex, self-reported health status, and medical history were recorded. Blood samples were obtained under fasting conditions and analyzed within 2 h of centrifugation. The blood chemistry studied was assessed by standard automated techniques, including Architect C 8000 auto-analyzer and Axsyme Third-Generation Immunoassay System (Abbott, Abbott Park, IL, USA). This analysis included indicators of liver function, glycometabolism, lipid metabolism, myocardial function, thyroid function, and inflammatory factors (Table S1).

## Statistical analysis

Descriptive statistics and Spearman correlation analysis were conducted using SPSS version 27.0 (SPSS Inc., Armonk, NY, USA). Continuous variables are presented as mean ± standard deviation, while categorical variables are presented as frequency (percentage). Then, assumption testing is employed to assess the normal distribution of continuous variables in the context of regression analysis. The normal distribution of continuous variables was assessed using the Kolmogorov–Smirnov and Shapiro–Wilk tests. The Mann–Whitney test was employed to compare the normal distribution between the two gender groups. The Kruskal-Wallis test was utilized to compare median differences among multiple blood type groups. $P$-values less than 0.05 in assumption testing were considered not consistent with the normal distribution. Spearman correlation analysis was performed to evaluate the relationship between biochemical indices and ferritin levels. Finally, age was treated as the independent variable, with gender and biochemical indicators associated with ferritin serving as control variables. Blood type was analyzed as a moderating variable to determine its effect on the relationship between age and ferritin levels. $P$-values less than 0.05 were considered statistically significant.

## RESULTS

### Descriptive statistical analysis of each biochemical index

We collected biochemical examination data from 3,843 individuals aged 40 and above who underwent health check-ups at Beijing Zhongguancun Hospital. A total of 2,960 healthy participants, with an average age of 53.96 ± 12.19 years, were included in this study, of which 900 (30.41%) were male. Among these participants, 859 (29.02%) had blood type O, 831 (28.07%) had type A, 978 (33.04%) had type B, and 292 (9.86%) had type AB (Table 1). There were individual differences in various biochemical indices, particularly in liver and kidney function, metabolic function, and myocardial damage. Notably, the mean ferritin level was 288.16, with a standard deviation of 99.89, indicating significant variability in ferritin levels among individuals. However, the median values of all indicators were close to the mean (Table 1), suggesting that the distribution of these indicators is relatively concentrated and suitable for further correlation and regression analyses. The flowchart of this study was shown in Fig. 2.

**Table 1  The characteristics of all variables.**

| Variable | N | Mean/(n, %) | Standard deviation | Median |
|---|---|---|---|---|
| Age | 2,960 | 53.959 | 12.185 | 51.000 |
| Gender | 2,960 | Male ($n = 900$, 30.41%) | | |
| | | Female ($n = 2060$, 69.59%) | | |
| Blood group | 2,960 | A ($n = 831$, 28.07%) | | |
| | | B ($n = 978$, 33.04%) | | |
| | | O ($n = 859$, 29.02%) | | |
| | | AB ($n = 292$, 9.86%) | | |
| Fer | 2,960 | 288.162 | 99.894 | 303.210 |
| ALT | 2,960 | 22.262 | 16.081 | 19.000 |
| AST | 2,960 | 22.375 | 13.315 | 20.300 |
| ALP | 2,960 | 73.943 | 25.553 | 82.510 |
| Cr | 2,960 | 68.404 | 22.250 | 65.000 |
| GGT | 2,960 | 30.733 | 28.747 | 22.000 |
| GLOB | 2,960 | 26.969 | 3.153 | 27.010 |
| A/G | 2,960 | 1.575 | 0.195 | 1.550 |
| Glu | 2,960 | 5.321 | 0.732 | 5.290 |
| sd LDL-C | 2,960 | 0.848 | 0.284 | 0.849 |
| LDL-C | 2,960 | 2.892 | 0.410 | 2.900 |
| HDL-C | 2,960 | 1.233 | 0.166 | 1.230 |
| Lpa | 2,960 | 205.518 | 117.043 | 224.090 |
| APOA1 | 2,960 | 1.319 | 0.194 | 1.290 |
| LDH | 2,960 | 234.456 | 41.973 | 253.960 |
| LDH-1 | 2,960 | 40.484 | 6.914 | 43.590 |
| CK | 2,960 | 112.398 | 38.566 | 119.700 |
| CK-MB | 2,960 | 15.736 | 4.454 | 17.560 |
| $\alpha$-HBDH | 2,960 | 168.106 | 25.968 | 178.310 |
| NT-pro BNP | 2,960 | 1,157.057 | 532.537 | 1,316.280 |
| TnT | 2,960 | 0.045 | 0.031 | 0.050 |
| Mb | 2,960 | 91.523 | 57.111 | 100.490 |
| TSH | 2,960 | 2.647 | 0.753 | 2.750 |
| FT3 | 2,960 | 4.301 | 0.385 | 4.240 |
| T3 | 2,960 | 1.510 | 0.158 | 1.490 |
| FT4 | 2,960 | 15.849 | 1.101 | 15.860 |
| IL-6 | 2,960 | 76.228 | 20.823 | 81.630 |

Notes.

N/n, number; Fer, ferritin; ALT, alanine aminotransferase; AST, aspartate aminotransferase; Cr, creatinine; GGT, gamma-glutamyltranspeptidase; ALP, alkaline phosphatase; GLOB, globulin; A/G, albumin/hlobulin; Glu, glucose; HDL-C, high density lipoprotein cholesterol; LDL-C, low density lipoprotein cholesterol; sd LDL-C, small and dense low density cholesterol; Lpa, lipoprotein a; APOA1, apolipoprotein A1; LDH, lactate dehydrogenase; LDH-1, lactate dehydrogenase isoenzyme; CK, creatine kinase; CK-MB, creatine kinase-MB isoenzyme activity; $\alpha$-HBDH, alpha hydroxybutyrate dehydrogenase; NT-pro BNP, N-terminal -B type natriuretic peptide precursor; TnT, troponin T; Mb, Myohemoglobin; TSH, thyroid stimulating hormone; FT3, free triiodothyronine; T3, triiodothyronine; FT4, free thyroxine; IL-6, interleukin-6.

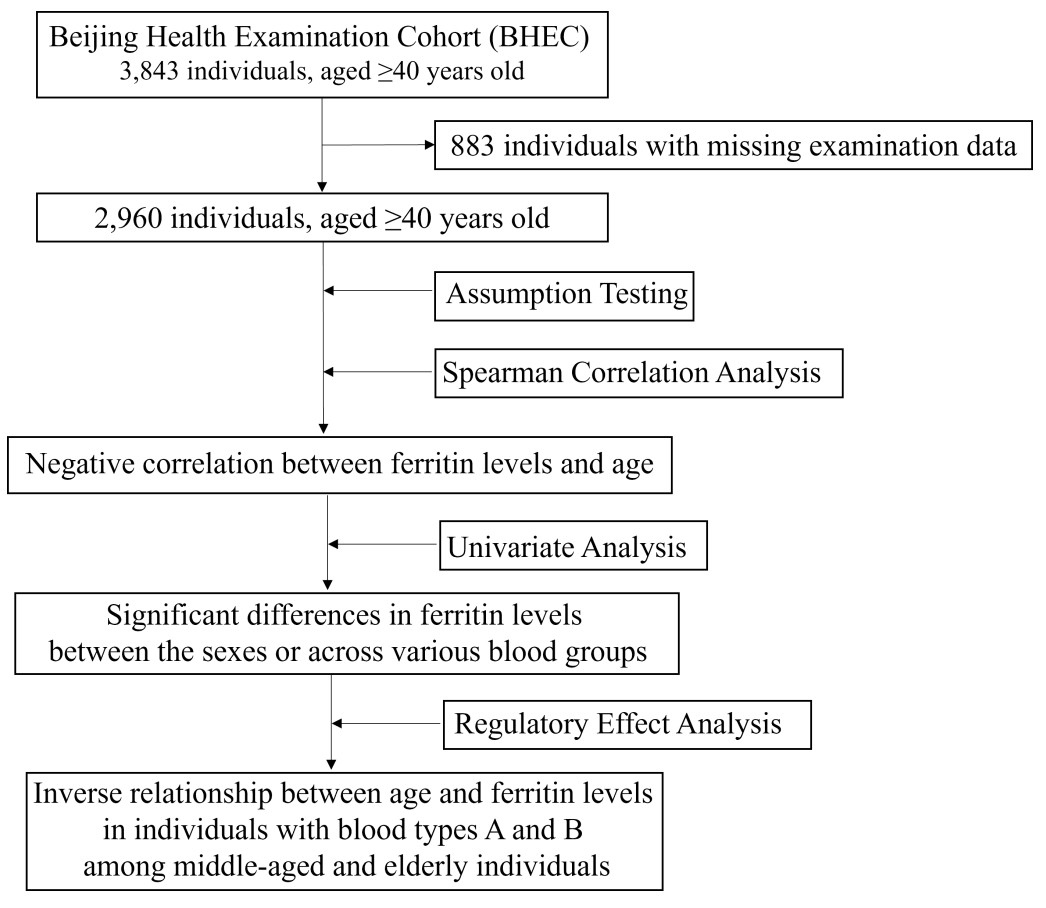

**Figure 2  The flowchart of this study.**

## Correlation and univariate analysis of ferritin and other indicators

Since all variables did not conform to a normal distribution ($p < 0.05$) (Table S2). Spearman correlation analysis was employed for continuous variables, while a non-parametric test was utilized for categorical variables.

The results indicated a negative correlation between ferritin levels and age ($\rho = -0.099$, $p < 0.001$), suggesting that ferritin levels tend to decrease as age increases (Table 2). Additionally, the difference in ferritin levels between men and women was statistically significant ($p = 0.005$) (Table 3), highlighting a notable disparity in ferritin levels between the sexes. Furthermore, there were statistically significant differences in ferritin levels across various blood groups ($p < 0.001$) (Table 4).

Alanine aminotransferase (ALT) ($\rho = 0.078$, $p < 0.001$), aspartate aminotransferase (AST) ($\rho = 0.039$, $p < 0.05$), γ-glutamyl transpeptidase (γ-GT) ($\rho = 0.074$, $p < 0.001$), creatinine (Cr) ($\rho = 0.105$, $p < 0.001$), alkaline phosphatase (ALP) ($\rho = 0.113$, $p < 0.001$), and lipoprotein(a) (LP(a)) ($\rho = 0.059$, $p < 0.01$) demonstrated significant positive correlations with ferritin (Table 2). This suggests that abnormalities in liver, bile, and renal function are associated with increased ferritin levels.

**Table 2  Spearman correlation analysis between continuous variables and ferritin.**

| Variables | Correlation coefficient |
|---|---|
| Age | $-0.099^{***}$ |
| ALT | $0.078^{***}$ |
| AST | $0.039^{*}$ |
| ALP | $0.113^{***}$ |
| Cr | $0.105^{***}$ |
| GGT | $0.074^{***}$ |
| GLOB | $0.026$ |
| A/G | $-0.035$ |
| Glu | $0.042^{*}$ |
| LDL-C | $0.017$ |
| HDL-C | $-0.067^{***}$ |
| sd LDL-C | $0.038^{*}$ |
| Lpa | $0.059^{**}$ |
| APOA1 | $-0.086^{***}$ |
| LDH | $0.224^{***}$ |
| LDH-1 | $0.221^{***}$ |
| CK | $0.156^{***}$ |
| CK-MB | $0.210^{***}$ |
| $\alpha$-HBDH | $0.195^{***}$ |
| NT-pro BNP | $0.189^{***}$ |
| TnT | $0.209^{***}$ |
| Mb | $0.207^{***}$ |
| TSH | $0.126^{***}$ |
| FT3 | $-0.174^{***}$ |
| T3 | $-0.168^{***}$ |
| FT4 | $0.027$ |
| IL-6 | $0.103^{***}$ |

**Notes.**

$^{*}$$p$-value $< 0.05$.
$^{**}$$p$-value $< 0.01$.
$^{***}$$p$-value $< 0.001$.

Fer, ferritin; ALT, alanine aminotransferase; AST, aspartate aminotransferase; Cr, creatinine; GGT, gamma-glutamyltranspeptidase; ALP, alkaline phosphatase; GLOB, globulin; A/G, albumin/hlobulin; Glu, glucose; HDL-C, high density lipoprotein cholesterol; LDL-C, low density lipoprotein cholesterol; sd LDL-C, small and dense low density cholesterol; Lpa, lipoprotein a; APOA1, apolipoprotein A1; LDH, lactate dehydrogenase; LDH-1, lactate dehydrogenase isoenzyme; CK, creatine kinase; CK-MB, creatine kinase-MB isoenzyme activity; $\alpha$-HBDH, alpha hydroxybutyrate dehydrogenase; NT-pro BNP, N-terminal -B type natriuretic peptide precursor; TnT, troponin T; Mb, Myohemoglobin; TSH, thyroid stimulating hormone; FT3, free triiodothyronine; T3, triiodothyronine; FT4, free thyroxine; IL-6, interleukin-6.

There was a significant negative correlation between apolipoprotein A1 (APOA1) and ferritin ($\rho = -0.086$, $p < 0.001$) (Table 2), suggesting that an increase in ferritin levels is associated with a decrease in high-density lipoprotein (HDL) metabolism.

There were significant positive correlations between ferritin and creatine kinase (CK) ($\rho = 0.156$, $p < 0.001$), lactate dehydrogenase (LDH) ($\rho = 0.224$, $p < 0.001$), and myoglobin (Mb) ($\rho = 0.207$, $p < 0.001$) (Table 2). These findings suggested that elevated ferritin levels may be closely associated with myocardial injury.

**Table 3 Univariate analysis of categorical variables (gender).**

| Gender | Median (P25, P75) | | Mann–Whitney test Statistic U value | Mann–Whitney test Statistic z value | *p*-value |
|---|---|---|---|---|---|
| | Male (*n* = 900) | Female (*n* = 2060) | | | |
| Fer | 301.520 (300.4, 303.6) | 302.210 (301.3, 305.4) | 869,207.500 | −2.786 | 0.005 |

**Table 4 Univariate analysis of categorical variables (blood group).**

| Blood type | Median (P25, P75) | | | | Kruskal–Wallis test Statistic H value | *p*-value |
|---|---|---|---|---|---|---|
| | O (*n* = 859) | A (*n* = 831) | B (*n* = 978) | AB (*n* = 292) | | |
| Fer | 303.050 (302.5, 303.8) | 303.080 (302.5, 303.7) | 305.070 (3,042.5, 305.7) | 306.040 (306.4, 307.6) | 24.780 | <0.001 |

Thyroid-stimulating hormone (TSH) ($\rho = 0.126$, $p < 0.001$) exhibited a positive correlation with ferritin, whereas free triiodothyronine (FT3) ($\rho = -0.174$, $p < 0.001$) and total triiodothyronine (TT3) ($\rho = -0.168$, $p < 0.001$) demonstrated negative correlations with ferritin (Table 2). These findings suggested a relationship between thyroid function and ferritin levels.

Interleukin-6 (IL-6) ($\rho = 0.103$, $p < 0.001$) exhibited a positive correlation with ferritin (Table 2), suggesting that ferritin levels are closely associated with the elevation of inflammatory factors.

Glucose (Glu) ($\rho = 0.042$, $p < 0.05$) and small and dense low-density lipoprotein cholesterol (sd LDL-C) ($\rho = 0.038$, $p < 0.05$) exhibited a weak positive correlation with ferritin (Table 2). This finding suggested that metabolic disorders may be associated with ferritin levels.

## The regulatory effect of blood group on ferritin levels with aging

We included indicators related to ferritin and analyzed the regulatory effects with age as the independent variable, gender and other biochemical indicators as control variables, and blood type as the moderating variable.

In Model 1, which analyzed the direct effects of age and several control variables (including gender, ALT, AST, creatinine, glucose, *etc.*) on ferritin levels, age exhibited a significant negative effect on ferritin levels (t = −2.340, $p = 0.019$). Specifically, as age increased, ferritin levels decreased significantly (Table 5). These results indicate that age is an independent factor influencing ferritin levels.

Model 2 incorporates blood type as a regulatory variable based on Model 1 to investigate whether different blood types influence the relationship between age and ferritin levels. The results indicated that blood group B ($p = 0.042$) and blood group AB ($p = 0.048$) had significant effects on ferritin levels compared to blood group O, while blood group A did not show a significant difference (Table 5). This suggests that blood groups B and AB may modulate the effect of age on ferritin levels to some extent.

**Table 5  The regulatory effect of blood group on the level of ferritin with aging.**

| | Model 1 | | | | | Model 2 | | | | | Model 3 | | | | |
|---|---|---|---|---|---|---|---|---|---|---|---|---|---|---|---|
| | B | Standard error | t | *p*-value | β | B | Standard error | t | *p*-value | β | B | Standard error | t | *p*-value | β |
| Age | −0.384 | 0.164 | −2.34 | 0.019 | −0.047 | −0.389 | 0.164 | −2.375 | 0.018 | −0.048 | −0.437 | 0.218 | −2.004 | 0.048 | −0.045 |
| Blood type O | - | | | | | | | | | | | | | | |
| Blood type A | | | | | | −4.872 | 4.652 | −1.047 | 0.295 | −0.022 | −4.758 | 4.640 | −1.026 | 0.305 | −0.021 |
| Blood type B | | | | | | −9.069 | 4.467 | −2.03 | 0.042 | −0.043 | −9.058 | 4.455 | −2.033 | 0.042 | −0.043 |
| Blood type AB | | | | | | −12.768 | 6.462 | −1.976 | 0.048 | −0.038 | −12.65 | 6.444 | −1.963 | 0.050 | −0.038 |
| Age*Blood type A | | | | | | | | | | | −1.143 | 0.388 | −2.943 | 0.003 | −0.072 |
| Age*Blood type B | | | | | | | | | | | −1.552 | 0.365 | −4.248 | $p < 0.001$ | −0.110 |
| Age*Blood type AB | | | | | | | | | | | −1.102 | 0.517 | −2.132 | 0.033 | −0.044 |
| Constant | 467.502 | 34.066 | 13.724 | $p < 0.001$ | – | 471.505 | 34.107 | 13.824 | $p < 0.001$ | – | 466.88 | 34.065 | 13.706 | $p < 0.001$ | – |
| Control variable | | Yes | | | | | Yes | | | | | Yes | | | |
| R² | | 0.101 | | | | | 0.103 | | | | | 0.109 | | | |
| Adjust R² | | 0.094 | | | | | 0.095 | | | | | 0.100 | | | |
| F value | | $F(24, 2935) = 13.781, p < 0.001$ | | | | | $F(27, 2932) = 12.482, p < 0.001$ | | | | | $F(30, 2929) = 11.929, p < 0.001$ | | | |

Model 3 further incorporated the interaction term of age and blood type to elucidate the specific regulatory effects of different blood types on the relationship between age and ferritin levels. The results indicated that various blood groups indeed play a significant regulatory role in this relationship. In blood group A ($p = 0.003$, $\beta = -0.072$) and blood group B ($p < 0.001$, $\beta = -0.110$), the negative impact of age on ferritin levels was more pronounced, demonstrating a stronger regulatory effect compared to other blood groups. In populations with blood group AB ($p = 0.033$, $\beta = -0.044$), the negative effect of age on ferritin was greater than that observed in blood group O, although the regulatory effect was relatively weak (Table 5).

## DISCUSSION

Ferritin is a vital protein complex responsible for storing excess cellular iron and plays a crucial role in various metabolic pathways, inflammatory processes, stress responses, and the pathogenesis of cancer and neurodegenerative diseases (*Fang et al., 2020*; *Sudarev et al., 2023*). In recent years, ferritin has been shown to play a central role in senescence, driving the senescence-associated secretory phenotype (SASP) through the induction of senescence and iron accumulation in senescent cells (*Maus et al., 2023*). A cohort study found that both low and high serum ferritin levels were associated with an increased risk of incident heart failure in the general population (*Silvestre et al., 2017*). A meta-analysis of 11 studies revealed that serum ferritin levels were higher in patients with non-alcoholic fatty liver disease (NAFLD) compared to healthy individuals and were associated with the risk of developing NAFLD (*Yan et al., 2023*). A longitudinal study involving 6,497 participants found that serum ferritin levels were positively and independently associated with the incidence of type 2 diabetes (T2D) and cerebrovascular disease (CEVD) (*Suárez-Ortegón et al., 2022*). In our study, we observed a significant correlation between ferritin levels and age, liver function, lipid metabolism, myocardial function, and other biochemical indicators, further suggesting the important role of ferritin in senescence and senescence-associated diseases (Table 2). Therefore, ferritin holds promise as a potential predictive biomarker for senescence-associated diseases.

The ABO blood groups have been associated with a variety of health conditions over the years. A genome-wide association study reported a significant correlation between the ABO locus and ferritin levels (*Benyamin et al., 2014*). An epidemiological study involving 30,595 participants from the Danish Blood Donor Study indicated that non-O blood group donors exhibited lower ferritin levels compared to those with blood group O (*Rigas et al., 2017*). However, the interaction between ABO blood groups and ferritin levels during senescence has not been fully elucidated. Our results suggest that the impact of age on ferritin levels varies significantly among different blood groups for the first time, particularly in individuals with blood groups A and B, where the negative correlation between age and ferritin levels is more pronounced (Table 5). The molecular mechanisms underlying the regulatory effect of blood group on ferritin levels in aging remain inadequately understood. However, it has been observed that with advancing age, there is a noted decline in testosterone levels accompanied by an increase in ferritin levels.

This phenomenon contributes to a reduction in iron absorption and utilization (*Sudarev et al., 2023*), thereby establishing a negative correlation between serum ferritin levels and age as observed in this study. Furthermore, variations in blood groups, attributable to differing capacities for erythropoiesis, result in distinct iron demands, which in turn exert a regulatory influence on serum iron levels (*Kronstein-Wiedemann et al., 2023*). Based on the above results, we recommended that middle-aged and elderly individuals with blood groups A and B consider iron supplementation.

### Strength and limitation

The strength of this study lies in its analysis of the regulatory effect of blood groups on ferritin levels in middle-aged and elderly individuals as a cohesive group. This approach helps to mitigate the loss of continuity associated with ageing that can occur when dividing participants into distinct age groups. Consequently, both the independent and dependent variables in this study are treated as continuous variables.

Our study has several limitations. First, the cross-sectional analysis of the data does not permit a causal assessment of the relationship under investigation. Second, while we have accounted for some confounding factors, our study lacks data on diet and lifestyle, which may influence ferritin levels. Third, although we included and analyzed some relevant biochemical indicators as control variables, the study did not comprehensively cover all biochemical indicators, and the results may be affected by other unmeasured biochemical factors. Therefore, we plan to incorporate questionnaires and follow-up assessments, as well as include additional biochemical indicators, to enhance the comprehensiveness and rigor of the study in the future.

## CONCLUSIONS

The ABO blood group appears to influence ferritin levels during ageing, particularly in individuals with blood groups A and B. In these groups, the negative correlation between age and ferritin levels is more pronounced among middle-aged and elderly individuals. The results indicate the possible advantages of administering targeted iron supplementation. Nevertheless, it is essential for future research to investigate the specific dosage-response relationship. The mechanisms by which ABO blood groups affect serum ferritin levels in senescence also require further investigation.

## ACKNOWLEDGEMENTS

The authors would like to express their gratitude to all the participants for their voluntary involvement, as well as to the community committee staff for their friendly support.

### Funding

This work was supported by the National Natural Science Foundation of China (grant number 82301776); State Key Laboratory Special Fund (grant number 2060204); Chinese

Academy of Medical Sciences Innovation Fund for Medical Sciences (grant number 2023-I2M-2-001). The funders had no role in study design, data collection and analysis, decision to publish, or preparation of the manuscript.

## Grant Disclosures
The following grant information was disclosed by the authors:
The National Natural Science Foundation of China: 82301776.
State Key Laboratory Special Fund: 2060204.
Chinese Academy of Medical Sciences Innovation Fund for Medical Sciences: 2023-I2M-2-001.

## Competing Interests
The authors declare there are no competing interests.

## Author Contributions
- Ni Xiaolin conceived and designed the experiments, performed the experiments, analyzed the data, prepared figures and/or tables, authored or reviewed drafts of the article, and approved the final draft.
- Fenghong Yao performed the experiments, prepared figures and/or tables, and approved the final draft.
- Mingkang Pan conceived and designed the experiments, authored or reviewed drafts of the article, and approved the final draft.

## Human Ethics
The following information was supplied relating to ethical approvals (i.e., approving body and any reference numbers):

The Ethics Committee of the Institute of Basic Medical Sciences, Chinese Academy of Medical Sciences.

## Data Availability
The raw data is available in the Supplementary File.

## Supplemental Information
Supplemental information for this article can be found online at http://dx.doi.org/10.7717/peerj.19281#supplemental-information.

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
