# Peer review of "The regulatory effect of blood group on ferritin levels in aging: a retrospective study"

_PeerJ, doi:10.7717/peerj.19281_

## Round 0.1 · original submission · Minor Revisions

Dear Author,

Please address all the minor comments of the reviewers and also add cross-sectional study to the title. Describe the assumption testing conducted for regression analysis and also include figures as relevant to ferritin level and age group. I hope you are able to make these changes timely and satisfactorily.

·

Basic reporting

This manuscript is unambiguous and well-written. However, a few corrections are required. In the case of the introduction, the rationale and hypothesis for the study were not clearly stated and the introduction was unnecessarily too long. Most of the citations were missing in this section.

Experimental design

The methodology is good. Also, for the tables in the manuscript, the footnotes did not have the full names of the abbreviations.

Validity of the findings

Results and conclusions were well-stated and statistically sound

·

Basic reporting

Abstract:
Evaluation;
The abstract provides a clear summary of the study, including methodology, results, and conclusion.
Comments/Suggestions;
The abstract effectively outlines the background and key findings but could be improved by stating the study design and statistical methods more explicitly. Additionally, highlighting the implications of the findings in the abstract would add value.

Introduction:
Provides a comprehensive background on ABO blood and serum ferritin.
Comments/Suggestions;
The introduction is well-written but could include more recent studies to support the discussion on the significance correlation between ABO blood and serum ferritin. Clarifying the research gap that this study addresses would make the introduction more impactful.

Experimental design

Methodology:
Evaluation;
Clearly defined retrospective study with appropriate study population and period. between July 2019 and July 2024
Comments/Suggestions;
The methodology section is thorough, detailing the study design and data collection methods. However, more details on how the standardized questionnaire was developed and validated would be beneficial. Additionally, the statistical analysis methods could be expanded upon for clarity.

Results:
Comments/Suggestions
The results are presented in a well-structured manner, using tables. However, the section would benefit from more figures.

Validity of the findings

Discussion:
Offers a good comparison with existing literature and highlights the clinical relevance of the findings.
Comments/Suggestions;
The discussion section is insightful but could further explore the potential mechanisms behind the observed outcomes.

Figures and Tables:
The tables are well-constructed and support the text effectively.
Comments/Suggestions;
Consider adding more visual aids (e.g., flowcharts, Figures). Ensure consistent formatting and labeling across all tables.

Additional comments

Conclusion:
Summarizes the key findings effectively
Comments/Suggestions;
The conclusion is clear and concise, but it would benefit from including more specific recommendations for clinical practice. Highlighting future research directions.

References:
While the references cover relevant studies, some sources (e.g., 1997,2009 and 2014) could be updated to reflect the latest research in the field

·

Basic reporting

• The Regulatory Effect of Blood Group on Ferritin Levels in Aging (Xiaolin Ni et al) is aligned with the journal's subject matter.
• This research demonstrates a clear, relevant, and significant research topic.
• The language could be improved including lines 56-58, 73, 74.

Experimental design

• The research was conducted rigorously and ethically, with the methods documented in sufficient detail.

Validity of the findings

• The research offered a novel viewpoint on how blood group type might influence or even regulate how aging affects ferritin levels in the blood. However, as the authors mentioned the study lacks data on some confounding factors such as diet and lifestyle that might have an important effect on ferritin levels.
• The data are examined statistically in detail and from various perspectives. The findings of the study are presented in a clear and organized manner.
• The discussion of results is brief, to the point, and easy for the reader to understand.
• The conclusion is clear and concise, effectively summarizing the study's key findings.

---

## Round 0.2 · accepted · Accept

I am happy with the changes made by the authors as per reviewers recommendations. I feel the manuscript is ready for publication.

·

Basic reporting

No comment

Experimental design

No comment

Validity of the findings

No comment

Additional comments

Please use SI units in Tables and Figures consistently, and follow standard procedures. For instance, "ml should be mL".

·

Basic reporting

no comment

Experimental design

no comment

Validity of the findings

no comment

Additional comments

All area covered